# Mitochondrial introgression by ancient admixture between two distant lacustrine fishes in Sulawesi Island

Mizuki Horoiwa[1¤], Ixchel F. Mandagi[2,3], Nobu Sutra[2,4], Javier Montenegro[2], Fadly Y. Tantu[5], Kawilarang W. A. Masengi[3], Atsushi J. Nagano[6], Junko Kusumi[7], Nina Yasuda[1], Kazunori Yamahira[2]*

1 Faculty of Agriculture, University of Miyazaki, Miyazaki, Japan, 2 Tropical Biosphere Research Center, University of the Ryukyus, Nishihara, Okinawa, Japan, 3 Faculty of Fisheries and Marine Science, Sam Ratulangi University, Manado, Indonesia, 4 Graduate School of Hasanuddin University, Makassar, Indonesia, 5 Faculty of Animal Husbandry and Fisheries, Tadulako University, Palu, Indonesia, 6 Faculty of Agriculture, Ryukoku University, Otsu, Japan, 7 Faculty of Social and Cultural Studies, Kyushu University, Fukuoka, Japan

¤ Current address: Tropical Biosphere Research Center, University of the Ryukyus, Nishihara, Okinawa, Japan
* yamahira@lab.u-ryukyu.ac.jp

**Data Availability Statement:** The ND2 sequences obtained in this study were deposited in DDBJ under accession numbers LC594688–LC594707. The ddRAD-seq reads were deposited in the DDBJ

## Abstract

Sulawesi, an island located in a biogeographical transition zone between Indomalaya and Australasia, is famous for its high levels of endemism. Ricefishes (family Adrianichthyidae) are an example of taxa that have uniquely diversified on this island. It was demonstrated that habitat fragmentation due to the Pliocene juxtaposition among tectonic subdivisions of this island was the primary factor that promoted their divergence; however, it is also equally probable that habitat fusions and resultant admixtures between phylogenetically distant species may have frequently occurred. Previous studies revealed that some individuals of *Oryzias sarasinorum* endemic to a tectonic lake in central Sulawesi have mitochondrial haplotypes that are similar to the haplotypes of *O. eversi*, which is a phylogenetically related but geologically distant (ca. 190 km apart) adrianichthyid endemic to a small fountain. In this study, we tested if this reflects ancient admixture of *O. eversi* and *O. sarasinorum*. Population genomic analyses of genome-wide single-nucleotide polymorphisms revealed that *O. eversi* and *O. sarasinorum* are substantially reproductively isolated from each other. Comparison of demographic models revealed that the models assuming ancient admixture from *O. eversi* to *O. sarasinorum* was more supported than the models assuming no admixture; this supported the idea that the *O. eversi*-like mitochondrial haplotype in *O. sarasinorum* was introgressed from *O. eversi*. This study is the first to demonstrate ancient admixture of lacustrine or pond organisms in Sulawesi beyond 100 km. The complex geological history of this island enabled such island-wide admixture of lacustrine organisms, which usually experience limited migration.

Sequence Read Archive under accession number DRA011122. Data files for phylogenetic analyses, population structure analyses, and fastsimcoal2 runs are provided as Supporting Information.

**Funding:** This study was supported by the Collaborative Research of Tropical Biosphere Research Center, University of the Ryukyus to JK (https://tbc.skr.u-ryukyu.ac.jp/cooperative-studies/), Core Research for Evolutional Science and Technology Grant Number JPMJCR20S2 (https://www.jst.go.jp/kisoken/crest/en/index.html), and JSPS KAKENHI Grant Numbers 26291093 and 17H01675 to KY (https://www.jsps.go.jp/english/e-grants/index.html). The funders had no role in study design, data collection and analysis, decision to publish, or preparation of the manuscript.

**Competing interests:** The authors have declared that no competing interests exist.

## Introduction

Sulawesi, an island located in a biogeographical transition zone between Indomalaya and Australasia, is famous for its high levels of endemism in both the terrestrial and freshwater fauna [1, 2]. This endemism indicates that these taxa diversified within the island. Sulawesi is composed of three major tectonic subdivisions, two of which originated in the Asian and Australian continental margins, and the other emerged by the orogeny due to tectonic collision between the two plates [3–6]. These three tectonic subdivisions have been juxtaposed with each other since the Pliocene (ca. 4 Mya) [7], and large portions of land have been uplifted over the last 2–3 Myr [8], which resulted in the current shape of Sulawesi. It was demonstrated that this complex geological history of the island may have largely affected the diversification of several Sulawesi endemic taxa (e.g., [9–12]).

Family Adrianichthyidae, commonly referred to as ricefishes or medaka, is one such taxon that has uniquely diversified on this island [12–14]. Previous studies revealed that adrianichthyids on this island are composed of six major clades (Fig 1) and demonstrated that divergence of these major clades largely reflected the tectonic activities of this island [12, 15]. In particular, habitat fragmentation due to the Pliocene juxtaposition among the tectonic subdivisions was the primary factor that drove divergence of the lacustrine lineages distributed in tectonic lakes of central Sulawesi [12]. However, it is less known how species or populations within each clade have diverged.

Within each major clade, individuals from a single species or population form a clade in most cases, which indicates that each species or population is phylogenetically distinct. However, there are several exceptions. For example, *O. sarasinorum*, *O. eversi*, and *O. dopingdopingensis*, which are endemic to Lake Lindu, Tilanga Fountain, and Doping-doping River, respectively, in western to central Sulawesi, form a major clade in mitochondrial phylogenies (named Clade 4 by [12, 15, 16]); however, two *O. sarasinorum* mitochondrial haplotypes are paraphyletic, and one of them forms a clade with *O. eversi* haplotypes (Fig 1). It remains unknown why these two mitochondrial haplotypes coexist in the *O. sarasinorum* population.

One possibility is mitochondrial introgression from *O. eversi* to *O. sarasinorum*. It is possible that the Pliocene juxtaposition of tectonic subdivisions of this island caused both fragmentations and fusions of tectonic lakes in central Sulawesi, which may have led to repeated isolations and admixtures of lacustrine organisms. Indeed, recent studies demonstrated ancient admixtures between lacustrine species that inhabit different lakes that are currently distant from each other in central Sulawesi [17]. It is possible that similar ancient admixture might have occurred between Lake Lindu and Tilanga Fountain that might have caused mitochondrial introgression.

Another possibility is incomplete lineage sorting (ILS). The topology of the mitochondrial tree might be incongruent with the topology of species tree because of ILS. ILS is very likely if *O. sarasinorum* and *O. eversi* are still young species that did not diverge a long time ago. To test if the paraphyly of *O. sarasinorum* mitochondrial haplotypes can be explained by ancient admixture versus ILS, comparisons of pre-defined models assuming different demographic histories by coalescent simulations (e.g., [18–20]) are very useful.

In this study, we first reconfirmed the composition of *O. sarasinorum* and *O. eversi* mitochondrial haplotypes by increasing the number of individuals examined. Second, we examined population genetic structures of the two species using genome-wide single nuclear polymorphisms (SNPs). Third, we tested whether ancient admixture or ILS was more likely to explain the coexistence of two mitochondrial haplotypes within *O. sarasinorum* by coalescent-based demographic comparisons. Based on these results, we demonstrated that the two distinct mitochondrial haplotypes within *O. sarasinorum* reflect historical introgressive hybridization.

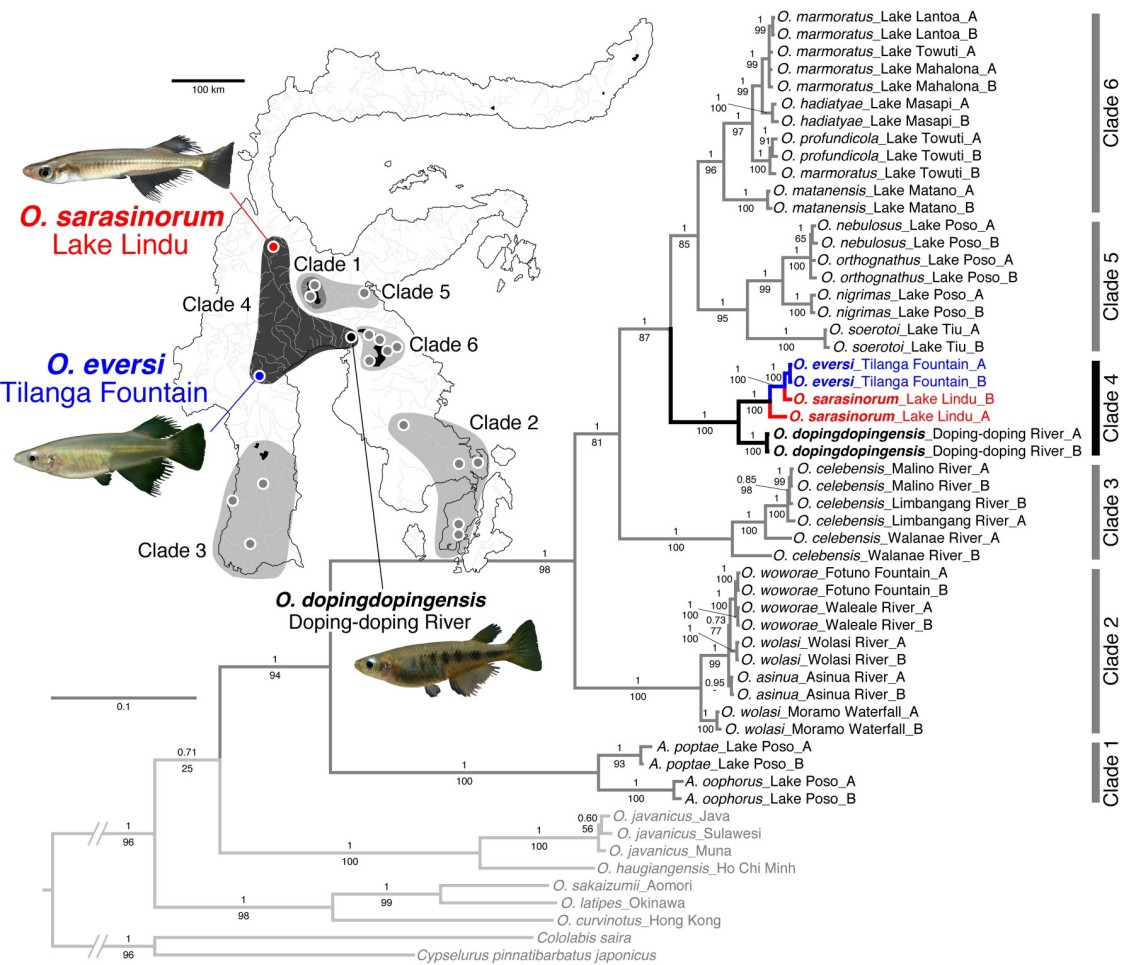

**Fig 1. Mitochondrial phylogeny of Sulawesi adrianichthyids and a map of Sulawesi with the distribution of the major lineages.**
The mitochondrial phylogeny was based on cyt b (1,141 bp) and ND2 (1,046 bp) (modified from [15]). Note that *Oryzias sarasinorum* and *O. eversi* are endemic to Lake Lindu and Tilanga Fountain, respectively, which are approximately 190 km apart. Numbers on branches are Bayesian posterior probabilities (top) and maximum likelihood bootstrap values (bottom). The scale bar indicates the number of substitutions per site.

## Materials and methods

### Field collections

Using a beach seine, we collected 10 juveniles each of *O. sarasinorum* and *O. eversi* from Lake Lindu (S01˚20′02″, E120˚03′09″) and Tilanga Fountain (S03˚02′07″, E119˚53′14″), respectively. Because *Oryzias* is not social, the possibility that the collected individuals were relatives with each other are probably quite low. They were preserved in 99% ethanol after being euthanized with MS-222. Total DNA was extracted from muscles of each of the 20 individuals using a DNeasy Blood & Tissue Kit (Qiagen, Hilden, Germany). Field collections were conducted with permission from the Ministry of Research, Technology, and Higher Education, Republic of Indonesia (research permit numbers 394/SIP/FRP/SM/XI/2014, 106/SIP/FRP/E5/Dit.KI/IV/2018, and 20/E5/E5.4/SIP.EXT/2019). We followed the Regulation for Animal Experiments at University of the Ryukyus for handling fishes, and all experiments were approved by the Animal Care Committee of University of the Ryukyus (2018099 and 2019084).

## Mitochondrial sequencing

The mitochondrial NADH dehydrogenase subunit 2 (ND2) gene was amplified for each of the 20 individuals (10 *O. sarasinorum* and 10 *O. eversi*) by PCR and Sanger sequenced using the methods and primers described by [12]. In addition, ND2 sequences of 10 *O. dopingdopingensis* individuals were retrieved from the DNA Data Bank of Japan (DDBJ) (LC551957–LC551966). *Oryzias dopingdopingensis* is a congener endemic to Doping-doping River in central Sulawesi (Fig 1). All sequences were aligned using the ClustalW option in MEGAX 10.1.8 [21], and the alignment was later manually corrected. We finally obtained 1,053 bp sequences of ND2 for the 30 individuals. Average pairwise genetic distances (p-distances) within and between *O. sarasinorum* and *O. eversi* were calculated using the MEGAX [21]. The ND2 sequences of the 10 *O. sarasinorum* and 10 *O. eversi* individuals were deposited in DDBJ under accession numbers LC594688–LC594707.

## ddRAD sequencing

For the *O. sarasinorum* and *O. eversi* individuals, genomic data were generated by ddRAD-seq [22], using restriction enzymes *Bgl*II and *Eco*RI (see [17] for details of library preparations). The library was sequenced with 50-bp single-end reads on an Illumina HiSeq 2500 system (Illumina, San Diego, USA) by Macrogen Japan Corporation (Kyoto, Japan). The sequencing reads were deposited in the DDBJ Sequence Read Archive under the accession number DRA011122 (S1 Table). In addition, raw ddRAD-seq reads (50-bp single-end) for each of the 10 *O. dopingdopingensis* individuals were obtained from the DDBJ Sequence Read Archive (DRA010303).

Sequence trimming was performed using Trimmomatic 0.32 [23] to remove adapter regions from the Illumina reads using the following settings: ILLUMICLIP:TruSeq3-SE. fa:2:30:10, LEADING:19, TRAILING: 19, SLIDINGWINDOW:30:20, and AVGQUAL:20, MINLEN:51. The remaining reads were mapped to a genome assembly of an *O. celebensis* individual [24]. Genotyping was conducted using the Stacks 1.48 software pipeline (*pstacks*, *cstacks*, and *sstacks*) [25, 26] with default settings except for the minimum to create a "stack," which was set to 10 reads (m = 10). The Stacks *populations* script was used to filter the loci that occurred in all three species (p = 3; i.e., *O. sarasinorum*, *O. eversi*, and *O. dopingdopingensis*) and in all individuals of each species (r = 1), i.e., no missing data was allowed. Loci that deviated from Hardy–Weinberg equilibrium (5% significance level) in one or more species were excluded from the dataset using VCFtools 0.1.13 [27]. Genotype outputs were created in VCF format for only the first SNP per locus (write_single_SNP), which resulted in 1,487 SNP sites. In addition, a PHYLIP file of concatenated sequences was created (phylip_var_all), which resulted in 3,790 loci with a total length of 193,290 bp. Similarly, we created genotype outputs among only *O. sarasinorum* and *O. eversi* (i.e., p = 2) in VCF format, which resulted in 4,703 RAD loci (S2 Table) that included 1,552 SNP sites. Among the 1,552 SNPs, 887 and 665 SNPs were transitional and transversional substitutions, respectively, and 290 SNPs were diagnostic, i.e., showing fixed differences between species.

## Phylogenetic analyses

A maximum-likelihood (ML) phylogeny among the 10 *O. sarasinorum*, 10 *O. eversi*, and 10 *O. dopingdopingensis* based on the 1,053-bp mitochondrial haplotypes was estimated with raxml-GUI 1.31 [28] using the codon-specific GTRGAMMAI model. *Oryzias dopingdopingensis* sequences were used as the outgroup, and bootstrap support values were calculated by a rapid bootstrap analysis of 1,000 bootstrap replicates. We also reconstructed a neighbor-joining (NJ) tree for the 193,290-bp concatenated RAD sequences using p-distances. Analysis was performed with MEGAX, and 1,000 bootstrap replicates were performed.

We also built a species tree based on the 1,487 RAD-seq SNPs using the Bayesian method implemented with SNAPP 1.4.1 [29]. Backward (U) and forward mutation rates (V) were estimated from the stationary allele frequencies in the data (U = 2.3066, V = 0.6384). Analysis was run using default priors with chainLength = 500,000 and storeEvery = 1,000. We discarded the first 10% of the trees as burn-in and visualized the posterior distribution of the remaining 450 trees as consensus trees using DensiTree 2.2.6 [30].

## Population structure analyses

We examined population structure within and among the three species with ADMIXTURE 1.3.0 [31] based on a PED file converted from the VCF file of the 1,487 RAD-seq SNPs using PLINK 1.90b4.6 [32]. ADMIXTURE was run for one to four clusters (i.e., K = 1–4). Statistical support for the different numbers of clusters was evaluated using the cross-validation technique implemented in ADMIXTURE. We also conducted principal component analyses using the R package SNPRelate 1.10.2 [33].

## Coalescence-based demographic inference

The demographic history of *O. sarasinorum* and *O. eversi* was inferred using fastsimcoal2 2.6.0.2 [19]. To better account for the complexity of multi-population models, we first compared five one-population models, which differ in population size change, separately for each species (S1 Fig) and chose the best-fit model for each species. One-hundred independent fastsimcoal2 runs with broad prior search ranges for each parameter were performed for each model using a one-dimensional site frequency spectrum created from the 4,703 RAD loci. We used a mutation rate of $3.5 \times 10^{-9}$ per site and generation for each run, which was estimated using a cichlid parent–offspring trio with whole-genome sequencing [34] to convert the inferred parameters into demographic units. The relative fit of each model to the data was evaluated by Akaike information criterion (AIC) after transforming the log10-likelihood values to ln-likelihoods. As a result, the model incorporating population growth in the past (Past-growth_model) had the highest support in both species (S3 Table).

Next, we designed three types of two-population models using the best one-population model (Fig 2, S4 Table). The first type assumed allopatric divergence without gene flow and admixture (ALD_model). If this model was supported, the paraphyly of *O. sarasinorum* mitochondrial haplotypes would indicate ILS. The second and third types assumed gene flow (DGF_model) and ancient admixture (ADM_models), respectively. The third type of models was further divided into two, one of which assumed direct admixture between *O. eversi* and *O. sarasinorum* (ADM1_model) and the other assumed admixture between a lineage diverged from *O. eversi* and *O. sarasinorum* (ADM2_model). If DGF_model or ADM_models are supported, the scenario of mitochondrial introgression is highly probable. One-hundred independent fastsimcoal2 runs were performed for each model using a two-dimensional joint minor allele site frequency spectrum created from the 4,703 RAD loci and the mutation rate of $3.5 \times 10^{-9}$ per site and generation. The relative fit of each model to the data was evaluated by AIC, as described above. For the best-fit model, 95% confidence intervals were calculated by parametric bootstrapping according to the program manual.

## Results

### Phylogeny and population structure

The mitochondrial ML phylogeny revealed two haplotype types within *O. sarasinorum* (Fig 3A), one of which was clustered with *O. eversi* haplotypes. The monophyly of this haplotype

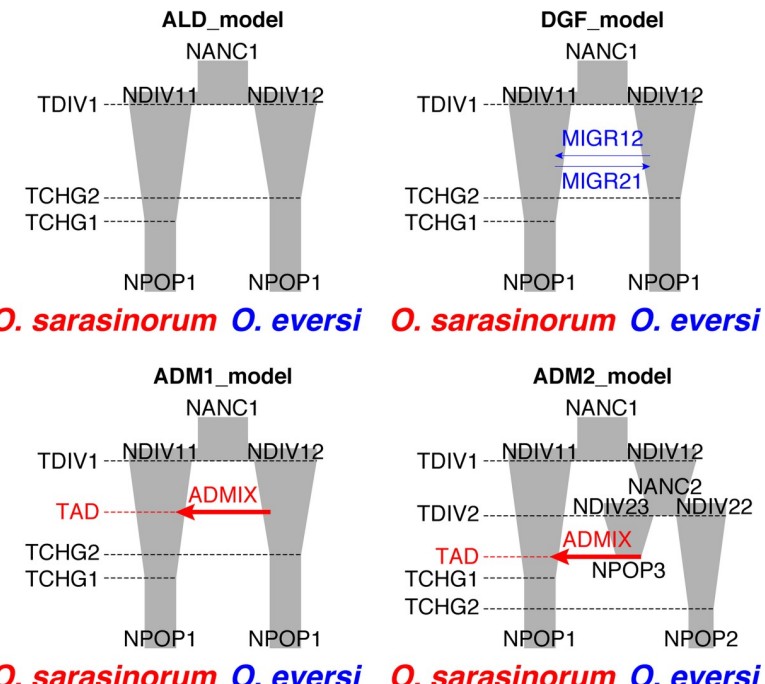

**Fig 2. Schematic illustration of two-population demographic models.** Note that growth was modeled as exponential and not linear as depicted here.

and *O. eversi* haplotypes had 100% ML bootstrap support. The other *O. sarasinorum* haplotypes formed a clade with 100% ML bootstrap support. Intraspecific average genetic distance was much higher in *O. sarasinorum* than in *O. eversi* (S5 Table).

In contrast, the nuclear NJ phylogeny based on the concatenated RAD sequences (193,290 bp) did not reveal two clusters within *O. sarasinorum* (Fig 3B). All *O. sarasinorum* individuals formed a clade with 99% bootstrap support. All *O. eversi* individuals also formed a clade with 99% bootstrap support. The species tree estimated by SNAPP also yielded the same topology (S2 Fig). In the posterior distribution of the species trees, all of the trees supported a topology consistent with the NJ tree.

ADMIXTURE analysis based on 1,487 SNPs revealed that the occurrence of three clusters (K = 3) had the highest support, and that *O. sarasinorum* and *O. eversi* were clearly separated (Fig 4). These two species were also separated from each other by the second principal component (PC2) in the principal component analysis (S3 Fig).

## Demographic model selection

The model assuming direct ancient admixture (ADM1_model) was best supported by the fastsimcoal2 runs (Table 1). In this model, the common ancestor of *O. sarasinorum* and *O. eversi* diverged approximately 85,000 (78,867–158,420) generations ago (Fig 5A, Table 2). Population size of *O. sarasinorum* and *O. eversi* was estimated to have grown and shrunk, respectively, after they diverged from each other. Approximately 7,700 (1,898–21,372) generations ago, *O. sarasinorum* experienced introgression from *O. eversi*. The ratio of *O. eversi* migrants to *O. sarasinorum* was estimated to be 2.3% (1.2–6.1%).

The model assuming admixture between a lineage diverged from *O. eversi* and *O. sarasinorum* (ADM2_model) was second best (Table 1). The time of admixture (TAD = ca. 6,000 generations ago) and the ratio of the migrants (ADMIX = 4.3%) were estimated to be similar

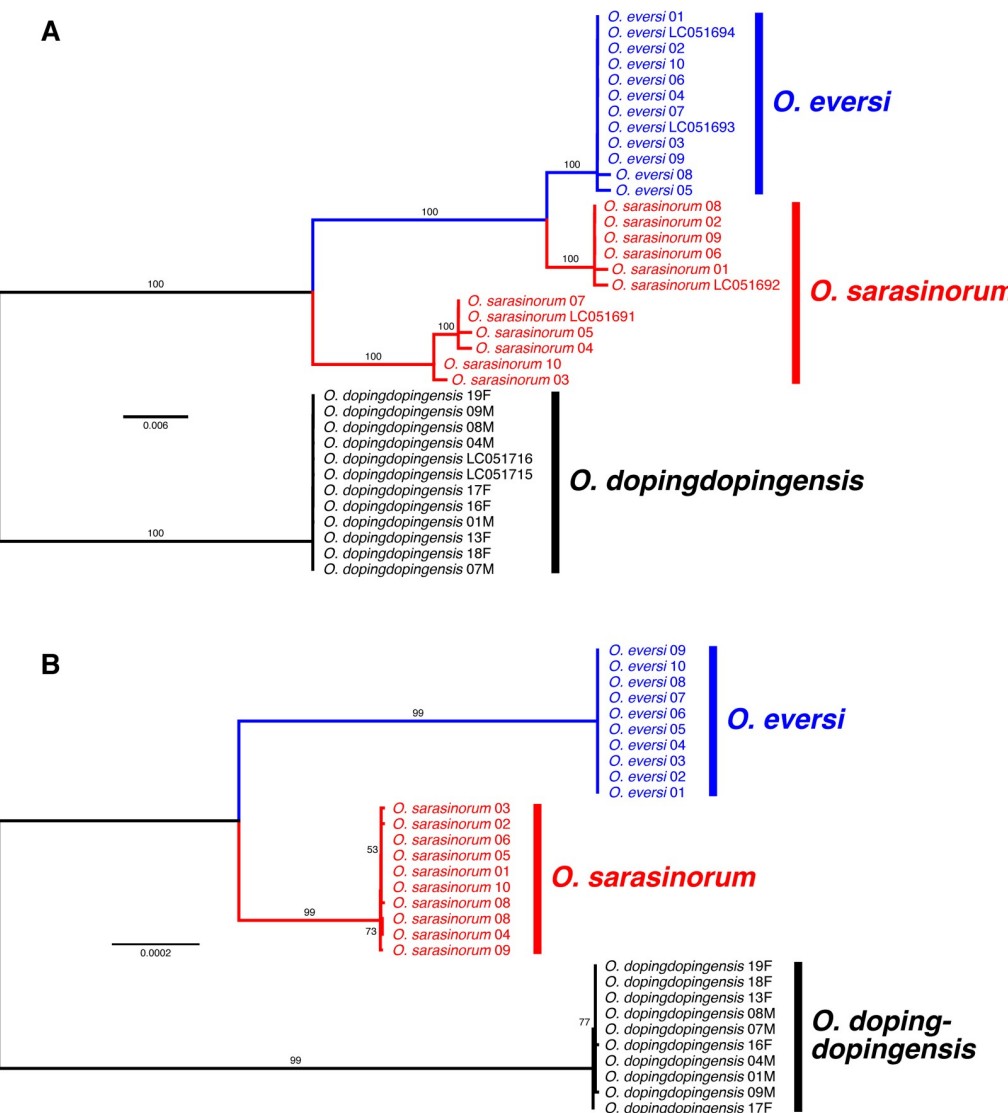

**Fig 3. Phylogenies of *Oryzias sarasinorum* and *O. eversi*.** (A) Maximum-likelihood phylogeny based on the 1,053-bp mitochondrial ND2 sequences and (B) neighbor-joining phylogeny based on the 193,290-bp concatenated RAD sequences. Numbers on branches are bootstrap values.

with those estimated by ADM1_model (S3 Table). These admixture models were much better supported than the model assuming gene flow (DGF_model) and the model assuming allopatric divergence with no gene flow and no admixture (ALD_model) (Table 1, Fig 5B, S6 Table).

## Discussion

### Ancient admixture and introgressive hybridization between the two distant lacustrine and pond species

The mitochondrial phylogeny in this study revealed that *O. sarasinorum* mitochondrial haplotypes were not monophyletic, and some haplotypes were clustered with *O. eversi* haplotypes. However, the nuclear phylogeny showed monophyly of the *O. sarasinorum* individuals, which were clearly separated from *O. eversi* individuals. The population structure analyses also

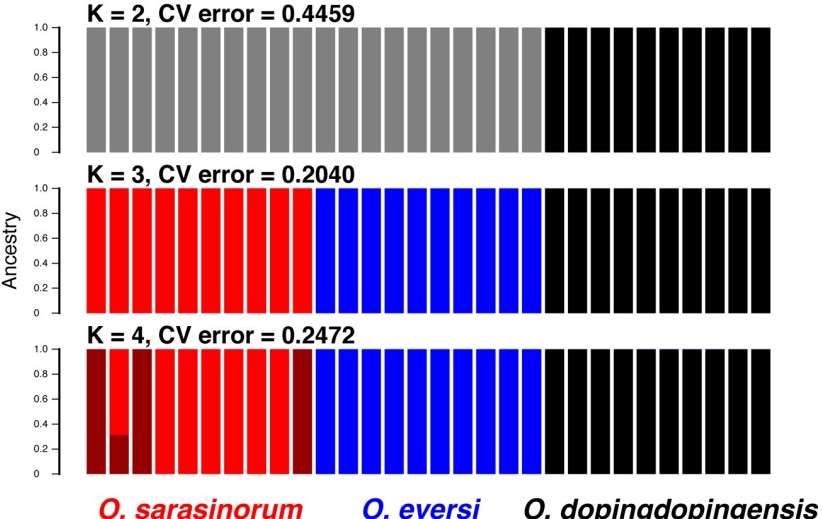

**Fig 4. ADMIXTURE results showing K = 2–4 genetic clusters.** Analysis was based on 1,487 SNPs among the three species.

revealed that *O. sarasinorum* and *O. eversi* were clearly distinct from each other. These findings indicate that the two species are currently reproductively isolated from each other. However, the coalescence-based demographic analyses supported the scenario that assumes ancient admixture from *O. eversi* to *O. sarasinorum*; this indicates that the *O. sarasinorum* mitochondrial haplotypes that are close to those of *O. eversi* reflect introgression from *O. eversi* to *O. sarasinorum* rather than ILS.

Lake Lindu and Tilanga Fountain are currently ca. 190 km from each other. It is thought that the common ancestor of lacustrine adrianichthyids (Clades 4–6 in Fig 1) endemic to tectonic lakes in central Sulawesi was distributed in a big lake or lake system until the Pliocene (ca. 4 Mya), but that it was later fragmented into several lakes or lake systems [12]. The sister relationship between *O. sarasinorum* and *O. eversi* indicates that there was a time when their common ancestor was isolated in a lake that was later divided into two smaller lakes: one is present-day Lake Lindu and the other Tilanga Fountain.

However, it is possible that the lake did not just undergo division. Some tectonic lakes and lake systems are known to have undergone repeated fragmentations and fusions, which caused repeated isolations and admixtures of lacustrine organisms [17]. It is probable that Lake Lindu and Tilanga Fountain were repeatedly connected to each other even after being divided. A long rift valley created by the action of the Palu–Koro fault system is located in the north–south direction between Lake Lindu and Tilanga Fountain [3, 8]; if there was a time when this rift valley was a rift valley lake, then Lake Lindu and Tilanga Fountain would not have been as isolated from each other as they are now. This scenario is quite likely, because the Plio-Pleistocene uplift of large portions of land [8] may have simultaneously changed river and lake systems on this island drastically.

**Table 1. Support for each two-population model.**

| Model | Number of parameters | log10-likelihood | Relative likelihood | ln-likelihood | AIC | Δ-AIC |
|---|---|---|---|---|---|---|
| ADM1 | 10 | −6,390.633 | — | −14,714.976 | 29,449.953 | — |
| ADM2 | 15 | −6,390.714 | $8.299 \times 10^{-1}$ | −14,715.163 | 29,460.326 | 10.373 |
| DGF | 10 | −6,407.783 | $7.079 \times 10^{-18}$ | −14,754.466 | 29,528.931 | 78.979 |
| ALD | 8 | −6,410.675 | $9.078 \times 10^{-21}$ | −14,761.125 | 29,538.249 | 88.297 |

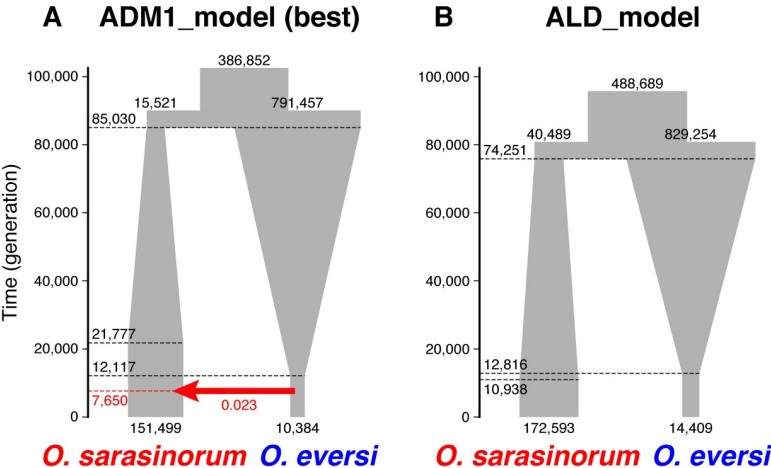

**Fig 5.** Schematic illustration of (A) ADM1_model (the best model) and (B) ALD_model estimated by fastsimcoal2 runs. The model is drawn to scale (time in generations) and population sizes; however, growth was modeled to be exponential and not linear as depicted here. The red arrow represents admixture.

Indeed, a fossil of an adrianichthyid species, †*Lithopoecilus brouweride*, was reported from this rift valley (Gimpoe Basin) in the Miocene (ca. 23.0–5.3 Mya) geological stratum [35, 36]. †*Lithopoecilus* is morphologically intermediate between *Oryzias* and *Adrianichthys*, a larger adrianichthyid genus [35], just like *O. sarasinorum* [37]. Although the exact generation time for these species remains unknown, assuming a generation time of 2 years as in [17], our coalescent-base demographic inference estimated that the divergence between *O. sarasinorum* and *O. eversi* was approximately 170,000 (158,000–317,000) years ago. Therefore, we think that †*L. brouweride* is the common ancestor of *O. sarasinorum* and *O. eversi*, but further examinations of this fossil species are necessary. The Miocene strata of Sulawesi consist of shallow-marine and terrestrial deposits [38], but this fossil specimen was contained in well-laminated mudstone [36] which should have deposited in lacustrine environments. This supports our view that there was a lake or lake system there until Pliocene. Either way, it is certain that there was an adrianichthyid in between Lake Lindu and Tilanga Fountain, which is currently land. It is therefore possible that the two species that are presently 190 km apart underwent historical admixture.

**Table 2. Confidence intervals for each parameter in the best model (ADM1_model).**

| Parameter | 95% Confidence interval |
|-----------|-------------------------|
| NPOP1 | 111,545–196,162 |
| NPOP2 | 10,972–21,694 |
| NDIV11 | 11,714–605,060 |
| NDIV12 | 203,803–1,570,705 |
| NANC1 | 238,169–612,834 |
| TCHG1 | 4,542–154,938 |
| TCHG2 | 4,820–32,347 |
| TDIV1 | 78,867–158,420 |
| TAD | 1,898–21,372 |
| ADMIX | 0.01269–0.06063 |

The 95% confidence intervals were obtained from nonparametric bootstrapping.

Assuming a generation time of 2 years, the age of the admixture between *O. eversi* and *O. sarasinorum* was estimated to be ca. 4,000–43,000 years ago. However, this estimate may be too young. The divergence between the *O. eversi* mitochondrial haplotypes and the *O. eversi*-like *O. sarasinorum* haplotypes (i.e., p-distance = 0.807%) would have occurred ca. 260,000–322,000 years ago assuming a substitution rate of 2.5%–3.1% per million years [39], which has been used for divergence-time estimation of Sulawesi adrianichthyids [2, 12, 17]. This discrepancy may indicate that the mutation rate used in the demographic inference (i.e., $3.5 \times 10^{-9}$ per site and generation) was too high.

### Endemism shaped by island-wide admixture

In summary, we demonstrated that *O. sarasinorum* and *O. eversi* have a history of being admixed even though they are currently distributed in geologically distant tectonic lakes. Ancient admixture within single lake systems or between adjacent lakes has been demonstrated from other lakes or lake systems in central Sulawesi not only in adrianichthyids [17] but also in other freshwater taxa [40–42]; however, this study is the first to demonstrate admixture beyond 100 km. It is the geological history of Sulawesi that enabled such an island-wide admixture event of lacustrine organisms, which usually experience limited migration. We also think that such repeated admixtures may have promoted diversification of this freshwater fish group and probably other freshwater taxa, because it has been recognized that hybridization facilitates rapid speciation and adaptive radiation (e.g., [43–45]). The high levels of endemism in many terrestrial and freshwater fauna on Sulawesi may have been shaped by repeated admixture between distant lineages caused by the complex geological history of this island.

## Supporting information

**S1 Fig. Schematic illustration of one-population demographic models.** Note that growth was modeled to be exponential and not linear as depicted here.
(TIF)

**S2 Fig. Species trees estimated by SNAPP based on 1,487 SNPs.** Thin lines represent individual species trees.
(TIF)

**S3 Fig. Principal component analysis of genetic variance based on 1,487 SNPs.**
(TIF)

**S1 Table. Sequencing reads deposited in the DDBJ Sequence Read Archive (accession number DRA011122).**
(DOCX)

**S2 Table. Genetic diversity of RAD locus.** S: number of segregating sites, H: number of haplotypes, Hd: haplotype diversity, π: nucleotide diversity, K: average number of nucleotide difference, Ho: per site observed heterozygosity averaged over samples, and Tajima's D. Each value represents the average among 4,703 loci. Calculations were performed by DnaSP 6.X.X.
(DOCX)

**S3 Table. Support for one-population models defined in S1 Fig.**
(DOCX)

**S4 Table. Explanation of each parameter used in the coalescent-based demographic inference.**
(DOCX)

**S5 Table. Intraspecific (diagonal) and interspecific (bottom left) average pairwise genetic distance (p-distance) based on the mitochondrial sequences.**
(DOCX)

**S6 Table. Inferred maximum-likelihood parameters for each model.**
(DOCX)

**S1 File. ND2 sequences (1,053 bp) used for the estimation of ML phylogeny.**
(FAS)

**S2 File. Concatenated RAD sequences (193,290 bp) used for the estimation of NJ phylogeny.**
(PHYLIP)

**S3 File. SNP data (1,487 SNPs) used for SNAPP and population structure analyses.**
(VCF)

**S4 File. RAD loci (4,703 loci) shared by *Oryzias sarasinorum* and *O. eversi*.**
(FA)

**S5 File. SNP data (1,552 SNPs) used to create the site frequency spectrum for fastsimcoal2.**
(VCF)

**S6 File. Site frequency spectra used for the fastsimcoal2 runs.**
(XLSX)

## Acknowledgments

We thank the Ministry of Research, Technology, and Higher Education, Republic of Indonesia (RISTEKDIKTI); the Faculty of Fisheries and Marine Science, Sam Ratulangi University; and the Faculty of Animal Husbandry and Fisheries, Tadulako University for the permit to conduct research in Sulawesi. Dr. Toshihiro Yamada, Osaka City University, kindly looked at the photo of †*Lithopoecilus brouweri* and gave us useful information about the environment where it fossilized. We also thank Dr. Mallory Eckstut from Edanz Group (https://en-author-services.edanz.com/ac) for editing a draft of this manuscript.

## Author Contributions

**Conceptualization:** Kazunori Yamahira.

**Data curation:** Javier Montenegro, Kazunori Yamahira.

**Formal analysis:** Mizuki Horoiwa, Nina Yasuda, Kazunori Yamahira.

**Funding acquisition:** Junko Kusumi, Kazunori Yamahira.

**Investigation:** Ixchel F. Mandagi, Nobu Sutra, Fadly Y. Tantu, Kawilarang W. A. Masengi, Atsushi J. Nagano, Kazunori Yamahira.

**Methodology:** Junko Kusumi, Kazunori Yamahira.

**Project administration:** Kazunori Yamahira.

**Supervision:** Kazunori Yamahira.

**Validation:** Kazunori Yamahira.

**Visualization:** Kazunori Yamahira.

**Writing – original draft:** Mizuki Horoiwa, Nina Yasuda, Kazunori Yamahira.

**Writing – review & editing:** Junko Kusumi.

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
