## [Decision Letter · Decision Letter 0]

9 Feb 2021

PONE-D-20-40353

Mitochondrial introgression by ancient admixture between two distant lacustrine fishes in Sulawesi Island

PLOS ONE

Dear Dr. Yamahira,

Thank you for submitting your manuscript to PLOS ONE. After careful consideration, we feel that it has merit but does not fully meet PLOS ONE’s publication criteria as it currently stands. Therefore, we invite you to submit a revised version of the manuscript that addresses the points raised during the review process.

We look forward to receiving your revised manuscript.

Kind regards,

Tzen-Yuh Chiang

Academic Editor

PLOS ONE

Journal Requirements:

Reviewers' comments:

Reviewer's Responses to Questions

**Comments to the Author**

1. Is the manuscript technically sound, and do the data support the conclusions?

Reviewer #1: Yes

Reviewer #2: Partly

2. Has the statistical analysis been performed appropriately and rigorously? 

Reviewer #1: Yes

Reviewer #2: I Don't Know

3. Have the authors made all data underlying the findings in their manuscript fully available?

Reviewer #1: Yes

Reviewer #2: Yes

4. Is the manuscript presented in an intelligible fashion and written in standard English?

Reviewer #1: Yes

Reviewer #2: Yes

5. Review Comments to the Author

Reviewer #1: Mitochondrial introgression by ancient admixture between two distant lacustrine fishes in Sulawasi Island

Mizuki Horoiwa et al.

The phylogenetic relationship between the different species of the Orysias genus was studied in a manuscript (2015) and it is clear that the mitochondrial DNA pattern between the two species Orysias sarasinorum (endemic Lake Lindu) and O. eversi (Tilanga fountain), which are approximately 190 km apart, was surprising (O. eversi mitochondrial haplotype in O. sarasinorum population). In this study, Horoiwa et al. focus on the evolutionary process responsible of the observed pattern. They test two evolutionary scenarios : incomplete lineage sorting versus introgressive hybridization between the two endemic species O. sarasinorum and O. eversi using the NADH dehydrogenase subunit 2 mitochondrial genes and 1552 SNP.

It is an interesting research based on species belonging to the genus of the well-known species model O. latipes. The final result is clear (hybridization in a “recent” past between the two species). The scientific question is clearly exposed and the results are based on a rigorous protocol and a good analysis.

Particular comments:

The figures in the pdf file are extremely difficult to read. I need to increase up to 200%

It will be interesting to introduce the number of each line in the manuscript.

Specific comments:

Introduction

“Another possibility is incomplete lineage sorting “: could the authors indicate the mean genetic distance between/within group for the two species based on cytochrome b. The information could be a little bit redundant with the Figure 1, but in my opinion this second hypothesis make sense only if the genetic divergence is low (<0.05 on cyt b in teleosteans, as observed in the tree figure 1) otherwise the hypothesis 1 (hybridization) is sufficient.

Material and Methods

Field collections

Page 5. How collected specimens using beach seine in one place, one time was representative to the diversity of the different species? The 10 specimens could belong to the same “shoal”/family and genetically highly related? Could you give an idea of the population size for the different species?

ddRAD sequencing

Page 5. “the reads sequencing were deposited in DDBJ Sequence Read Archive”. It is a good practice, I appreciate.

Page 5. Could you add the two restriction enzymes in the text: BglII and EcoRI

Page 5. Could you indicate the quality score Q of the reads that you used, Q>30?

Page 5. Could you indicate the number of raw reads for each specimen?

Page 5. I understand that the authors prefer to use a calibrated pipeline, however we have now the version 2.55 (https://catchenlab.life.illinois.edu/stacks/).

Page 6. “Loci deviated from Hardy-Weinberg equilibrium… “ It will be interesting to give some diversity indices such observed heterozygozity, private alleles. Considering that 10 specimens were collected, we need to have an idea about the polymorphism of the sampled specimen.

Page 6. Considering the 1,552 SNP sites could you indicated the % of missing data for each specimen, within species and between species.

How many SNP are diagnostic between the two species (as an example “A” fixed for O. sarasinorum and “G” fixed for O. eversi) ?

Could you indicate the number of transition and the number of transversion for the 1,552 SNP.

Phylogenetic analyses/population structure/Coalescence based demographic inference

This part is well written, with numerous information on the parameters used and tested hypotheses.

Page 7. “We used a synonymous substitution rate of 3.5 10 x 10-9 per site per generation” Why synonymous ? The mapping of the SNP was done on coding sequence only ?

Is it possible to use DIYABC (ref 18) based on various prior distributions to estimate the posterior distribution of the substitution rate (median of the posterior distribution)?

Results

Depending to the % of missing data, it will be important to test the impact on the population structure and Coalescence based demographic inference (i.e. if the % is higher than 30%).

In my opinion if the % of missing data is lower than 30% it is not necessary.

Discussion

Page 11. Assuming the generation time… was to high. I agree with the authors, however it is important to have a better idea of the substitution rate parameter (and to have an idea of the impact of missing data, if missing data there are).

Another question is how genetically highly related specimen for each species could impact the analyses and the conclusion?

Reviewer #2: This is an interesting case. The team of authors focuses on a previously known case of paraphyly in mitochondrial haplotypes of Oryzias sarasinorum, ricefish endemic to Lake Lindu in Sulawesi. They extend the mitochondrial data, show that the two species analyzed (O. sarasinorum, O. eversi) are respectively monophyletic in their nuclear genomes compared to O. dopingdopingensis, and suggest possible scenarios that might explain the case.

While I consider analyzing the case of paraphyly very interesting, the manuscript suffers from substantial discrepancies among actual findings and conclusions, and also from limited coverage of the relevant literature.

Major issues:

1) The scenario of “three major tectonic subdivisions” (Introduction; cited are Hall 2009, 2011, Sparkman & Hall 2010) does not reflect the current state of knowledge: Substantial revisions of that concept have been proposed by Hall & Nugraha 2017, and Nugraha & Hall (2018). The authors cite one of these more recent references latter in the Discussion (as Nagrahaa & Hall 2018), however without incorporating or discussing the core framework of paleo-islands and expansions. As the geographic scenario proposed here rests on an apparently outdated geological background, the evolutionary implications proposed here also appear not valid.

2) Likewise, the assumption of a “lake or lake system” scenario (Discussion), and its possible fragmentation, is largely speculative, based not on geological or limnological data, but exclusively on a genomic study on ricefish speciation in Lake Poso, published by the same group of authors (Sutra et al. 2019). Core presumptions of the discussion are not covered by published studies or data, this is not acceptable.

3) Tilanga Fountain (the site where Oryzias eversi is endemic) is not a lake (Abstract, Discussion), it’s a 4 m deep and 10 m wide karst pond, an extension of a stream (see the original description of O. eversi). The present manuscript implies hybridization among lake endemics, which is not the case. Oryzias eversi should accordingly also not be called a “lacustrine species” (first header of the Discussion).

4) The selection of references is strongly biased towards the own work of the team of authors. It ignores previous findings on mitochondrial introgression and hybridization in Sulawesi lake fishes (e.g., Herder et al. 2006 Proc Roy Soc B, Schwarzer et al. Hydrobiologia 2008), as well as other work on Sulawesi ricefishes (to be found in the eLIFE review by Schwarzer & Hilgers 2019). A sound interpretation should take the relevant spectrum of reference into consideration.

In sum, the main line of discussion appears not valid when considering the points raised above. What remains is an interesting case of past genetic exchange among two populations of fishes nowadays separated by a distance of ca. 190 km, that is in need for plausible explanation.

6. PLOS authors have the option to publish the peer review history of their article (what does this mean?). If published, this will include your full peer review and any attached files.

Reviewer #1: **Yes: **Andre GILLES

Reviewer #2: No

---

## [Author Response · Author response to Decision Letter 0]

21 Feb 2021

Response to Reviewers

Reviewer #1: Mitochondrial introgression by ancient admixture between two distant lacustrine fishes in Sulawasi Island

Mizuki Horoiwa et al.

The phylogenetic relationship between the different species of the Orysias genus was studied in a manuscript (2015) and it is clear that the mitochondrial DNA pattern between the two species Orysias sarasinorum (endemic Lake Lindu) and O. eversi (Tilanga fountain), which are approximately 190 km apart, was surprising (O. eversi mitochondrial haplotype in O. sarasinorum population). In this study, Horoiwa et al. focus on the evolutionary process responsible of the observed pattern. They test two evolutionary scenarios : incomplete lineage sorting versus introgressive hybridization between the two endemic species O. sarasinorum and O. eversi using the NADH dehydrogenase subunit 2 mitochondrial genes and 1552 SNP.

It is an interesting research based on species belonging to the genus of the well-known species model O. latipes. The final result is clear (hybridization in a “recent” past between the two species). The scientific question is clearly exposed and the results are based on a rigorous protocol and a good analysis.

Particular comments:

1) The figures in the pdf file are extremely difficult to read. I need to increase up to 200%.

2) It will be interesting to introduce the number of each line in the manuscript.

We collected them accordingly.

Specific comments:

Introduction

3) “Another possibility is incomplete lineage sorting”: could the authors indicate the mean genetic distance between/within group for the two species based on cytochrome b. The information could be a little bit redundant with the Figure 1, but in my opinion this second hypothesis make sense only if the genetic divergence is low (<0.05 on cyt b in teleosteans, as observed in the tree figure 1) otherwise the hypothesis 1 (hybridization) is sufficient.

We calculated intra- and interspecific pairwise genetic distance (p-distance) based on the mitochondrial sequences, as suggested (see L161–162, L285–286, and S5 Table).

Material and Methods

Field collections

4) Page 5. How collected specimens using beach seine in one place, one time was representative to the diversity of the different species? The 10 specimens could belong to the same “shoal”/family and genetically highly related? Could you give an idea of the population size for the different species?

The demographic inference by fastsimcoal2 revealed that the population size of O. sarasinorum and that of O. eversi were estimated about 40,700–86,300 and 2,500–7,200, respectively (see Fig 5 and S6 Table). These estimations are intuitive; the former inhabits in a large lake (34.5 km2) while the latter in a small fountain (~20 m wide). Because Oryzias is not social, we guess that the possibility that the collected individuals were relatives with each other are low. We mentioned this in the revised manuscript (L138–142).

ddRAD sequencing

5) Page 5. “the reads sequencing were deposited in DDBJ Sequence Read Archive”. It is a good practice, I appreciate.

We also appreciate this comment.

6) Page 5. Could you add the two restriction enzymes in the text: BglII and EcoRI.

Yes, they were BglII and EcoRI. We added the information of the restriction enzymes (L170).

7) Page 5. Could you indicate the quality score Q of the reads that you used, Q>30?

It was Q>20. We added more information about the filtering of raw reads (L178–181).

8) Page 5. Could you indicate the number of raw reads for each specimen?

We indicated the number of raw reads in S1 Table.

9) Page 5. I understand that the authors prefer to use a calibrated pipeline, however we have now the version 2.55 (https://catchenlab.life.illinois.edu/stacks/).

We appreciate this updated information. We had no problem using the calibrated pipeline.

10) Page 6. “Loci deviated from Hardy-Weinberg equilibrium… “It will be interesting to give some diversity indices such observed heterozygozity, private alleles. Considering that 10 specimens were collected, we need to have an idea about the polymorphism of the sampled specimen.

We calculated those indices for each species (S2 Table).

11) Page 6. Considering the 1,552 SNP sites could you indicated the % of missing data for each specimen, within species and between species.

No missing data was allowed in the present dataset. We clarified this point in the revised manuscript (L187–188).

12) How many SNP are diagnostic between the two species (as an example “A” fixed for O. sarasinorum and “G” fixed for O. eversi) ?

Among the 1,552 SNPs, 290 SNPs were diagnostic. We mentioned it in the revised manuscript (L206–207).

13) Could you indicate the number of transition and the number of transversion for the 1,552 SNP.

Among the 1,552 SNPs, 887 and 665 SNPs were transitional and transversional substitutions, respectively. We mentioned it in the revised manuscript (L195).

Phylogenetic analyses/population structure/Coalescence based demographic inference

14) This part is well written, with numerous information on the parameters used and tested hypotheses.

We appreciate this comment.

15) Page 7. “We used a synonymous substitution rate of 3.5 10 x 10-9 per site per generation” Why synonymous ? The mapping of the SNP was done on coding sequence only ?

It was our mistake. It was not “a synonymous substitution rate” but “a mutation rate”. We collected it accordingly (L248 and L267). This is the de novo mutation rate in a cichlid parent–offspring trio obtained from whole-genome sequencing data (Malinsky et al. 2018).

16) Is it possible to use DIYABC (ref 18) based on various prior distributions to estimate the posterior distribution of the substitution rate (median of the posterior distribution)?

We appreciate this suggestion. However, we would like to estimate the mutation rate of this group not by model approaches but by empirical observations. Indeed, we are planning to conduct a laboratory experiment to estimate the mutation rate of this group using parent–offspring trios. We hope that more plausible time estimations will be possible soon, when that mutation rate becomes available.

17) Depending to the % of missing data, it will be important to test the impact on the population structure and Coalescence based demographic inference (i.e. if the % is higher than 30%).

In my opinion if the % of missing data is lower than 30% it is not necessary.

As above, no missing data was allowed.

Discussion

18) Page 11. Assuming the generation time… was to high. I agree with the authors, however it is important to have a better idea of the substitution rate parameter (and to have an idea of the impact of missing data, if missing data there are).

As above, we are planning to conduct a laboratory experiment to estimate the mutation rate of this group using parent–offspring trios. We hope that more plausible time estimations will be possible soon, when that mutation rate becomes available.

19) Another question is how genetically highly related specimen for each species could impact the analyses and the conclusion?

As we mentioned above, the population size of O. sarasinorum and that of O. eversi were estimated about 40,700–86,300 and 2,500–7,200, respectively (see Fig 5 and S6 Table). We guess that the possibility that the collected individuals were relatives with each other are low, because Oryzias is not social.

Reviewer #2: This is an interesting case. The team of authors focuses on a previously known case of paraphyly in mitochondrial haplotypes of Oryzias sarasinorum, ricefish endemic to Lake Lindu in Sulawesi. They extend the mitochondrial data, show that the two species analyzed (O. sarasinorum, O. eversi) are respectively monophyletic in their nuclear genomes compared to O. dopingdopingensis, and suggest possible scenarios that might explain the case.

While I consider analyzing the case of paraphyly very interesting, the manuscript suffers from substantial discrepancies among actual findings and conclusions, and also from limited coverage of the relevant literature.

Major issues:

1) The scenario of “three major tectonic subdivisions” (Introduction; cited are Hall 2009, 2011, Sparkman & Hall 2010) does not reflect the current state of knowledge: Substantial revisions of that concept have been proposed by Hall & Nugraha 2017, and Nugraha & Hall (2018). The authors cite one of these more recent references latter in the Discussion (as Nagrahaa & Hall 2018), however without incorporating or discussing the core framework of paleo-islands and expansions. As the geographic scenario proposed here rests on an apparently outdated geological background, the evolutionary implications proposed here also appear not valid.

We appreciate this essential suggestion. We cited Nugraha and Hall (2018) (we could not find Hall and Nugraha 2017) in the Introduction section and mentioned that large portions of land have been uplifted since Pliocene (over the last 2–3 Myr) based on their findings (L62).

2) Likewise, the assumption of a “lake or lake system” scenario (Discussion), and its possible fragmentation, is largely speculative, based not on geological or limnological data, but exclusively on a genomic study on ricefish speciation in Lake Poso, published by the same group of authors (Sutra et al. 2019). Core presumptions of the discussion are not covered by published studies or data, this is not acceptable.

We discussed in the Discussion section that the “lake or lake system” scenario is quite likely, because the Plio-Pleistocene uplift of large portions of land, which was demonstrated by Nugraha and Hall (2018), may have simultaneously changed river and lake systems on this island drastically (L369–371). We also think that the fossil species (†Lithopoecilus brouweride) is a strong support to this scenario from geology.

3) Tilanga Fountain (the site where Oryzias eversi is endemic) is not a lake (Abstract, Discussion), it’s a 4 m deep and 10 m wide karst pond, an extension of a stream (see the original description of O. eversi). The present manuscript implies hybridization among lake endemics, which is not the case. Oryzias eversi should accordingly also not be called a “lacustrine species” (first header of the Discussion).

We went to Tilanga Fountain twice, and we found that this fountain is not an extension of a stream, that is, we found no surface connection with a stream, and that there is no water current like in a river. In other words, this fountain is completely isolated at least from any surface water systems, although we found water gushing out from underground, indicating an underground water system. That’s why we call O. eversi a “lacustrine species”, as an antonym of a “riverine species”, in this study. We clearly mentioned this in the revised manuscript (L138–142).

4) The selection of references is strongly biased towards the own work of the team of authors. It ignores previous findings on mitochondrial introgression and hybridization in Sulawesi lake fishes (e.g., Herder et al. 2006 Proc Roy Soc B, Schwarzer et al. Hydrobiologia 2008), as well as other work on Sulawesi ricefishes (to be found in the eLIFE review by Schwarzer & Hilgers 2019). A sound interpretation should take the relevant spectrum of reference into consideration.

We cited those non-adrianichthyid studies as other examples of mitochondrial introgression in the Discussion section (L396–407). We also included Hilgers and Schwarzer (2019) in the Introduction section as a study focusing on the adrianichthyid diversity on Sulawesi (L66–67).

In sum, the main line of discussion appears not valid when considering the points raised above. What remains is an interesting case of past genetic exchange among two populations of fishes nowadays separated by a distance of ca. 190 km, that is in need for plausible explanation.

---

## [Decision Letter · Decision Letter 1]

16 Mar 2021

PONE-D-20-40353R1

Mitochondrial introgression by ancient admixture between two distant lacustrine fishes in Sulawesi Island

PLOS ONE

Dear Dr. Yamahira,

Thank you for submitting your manuscript to PLOS ONE. After careful consideration, we feel that it has merit but does not fully meet PLOS ONE’s publication criteria as it currently stands. Therefore, we invite you to submit a revised version of the manuscript that addresses the points raised during the review process.

We look forward to receiving your revised manuscript.

Kind regards,

Tzen-Yuh Chiang

Academic Editor

PLOS ONE

Reviewers' comments:

Reviewer's Responses to Questions

**Comments to the Author**

1. If the authors have adequately addressed your comments raised in a previous round of review and you feel that this manuscript is now acceptable for publication, you may indicate that here to bypass the “Comments to the Author” section, enter your conflict of interest statement in the “Confidential to Editor” section, and submit your "Accept" recommendation.

Reviewer #1: All comments have been addressed

Reviewer #2: (No Response)

2. Is the manuscript technically sound, and do the data support the conclusions?

Reviewer #1: Yes

Reviewer #2: No

3. Has the statistical analysis been performed appropriately and rigorously? 

Reviewer #1: Yes

Reviewer #2: N/A

4. Have the authors made all data underlying the findings in their manuscript fully available?

Reviewer #1: Yes

Reviewer #2: Yes

5. Is the manuscript presented in an intelligible fashion and written in standard English?

Reviewer #1: Yes

Reviewer #2: Yes

6. Review Comments to the Author

Reviewer #1: (No Response)

Reviewer #2: In the revision, the team of authors addressed the issues raised to some extent. I appreciate that they incorporated additional literature on ricefishes, and also on the geology.

The scenario used for explaining the genetic results remains in my view still simply not justified. I see no independent evidence at all for the existence of the major lake, or a series of lakes, that is proposed as basis for the interpretation of the present results:

1) There is to the best of my knowledge no geological evidence that suggests the existence of a “big lake or lake system until the Pliocene” (l. 311-312), the same applied to the hypothesis of “several lakes or lake systems” (l. 312-313). The authors cite a phylogenetic paper on ricefishes from their own group for justifying this statement, but still fail in providing independent evidence. This is relevant, as the interpretation of the results largely collapses, once this speculation is removed.

2) The assumption that the fossil “…†Lithopoecilus brouweri is the common ancestor of O. sarasinorum and O. eversi…” (l. 335-336) is likewise pure speculation. The morphology of this fossil fish remains largely unclear: Frickhinger 1991 (cited here) illustrates the fossil, but gives little more information than that this is a small and slender fish with large eyes and a pointed head; Parenti 2008 (the second source cited here) says that she did not see the specimen, but that she sees no reason to contradict Beaufort 1934, who said that it is intermediate between Oryzias and Adrianichthys. In sum, the knowledge of †Lithopoecilus brouweri morphology is very limited. It can be said that a ricefish that was considered in 1934 being similar to other Sulawesi ricefishes occurred in the area in the Miocene – that’s it. Anything else would require new studies of this fossil specimen.

Further, I do not share the view that tiny Tilanga fountain is an environment that is to be termed “lacustrine”. Lacustrine refers to a “lake environment”, whereas the Tilanga fountain is a tiny pool, connected to groundwater, with a small overflow. The point I raise is however less that of the terminology, it questions if the habitat is from its properties somehow comparable to “real” lakes, such as Lake Lindu: Substantial and long-lived, largely stagnant waters. To my understanding, there is also no evidence that the Tilanga fountain can be seen as a leftover of a lake – its simply a fountain, and we do not know more about that habitat so far.

Having raised these points of criticism, I would like to express that I would in fact like seeing these interesting genetic results published, but with an interpretation that is based on the existing body of evidence. I feel that the idea of interconnection of the two sites, Tilanga and Lindu, by a lake or several lakes, could be proposed as one possible scenario – stating clearly that there is so far no further evidence available. And I would expect a critical discussion, including alternative scenarios. If this is done, the present paper will also remain valid if it should turn out later, that that there was in fact no lake connection among Tilanga and Lindu.

The same applies to the Tilanga habitat: Terming this “lacustrine” raises the expectation that these are two populations of lake fish, which is not the case. Saying openly that this is a “fountain” would appear to me even more interesting, and clearly correct.

7. PLOS authors have the option to publish the peer review history of their article (what does this mean?). If published, this will include your full peer review and any attached files.

Reviewer #1: No

Reviewer #2: No

---

## [Author Response · Author response to Decision Letter 1]

19 Mar 2021

Response to reviewer’s comments

1) There is to the best of my knowledge no geological evidence that suggests the existence of a “big lake or lake system until the Pliocene” (l. 311-312), the same applied to the hypothesis of “several lakes or lake systems” (l. 312-313). The authors cite a phylogenetic paper on ricefishes from their own group for justifying this statement, but still fail in providing independent evidence. This is relevant, as the interpretation of the results largely collapses, once this speculation is removed.

We showed a photo of †Lithopoecilus brouweri to a paleontologist (Dr. Toshihiro Yamada, Osaka City University) to consult him about the environment where it fossilized. According to him, this fossil specimen was contained in well-laminated mudstone which should have deposited in lacustrine environments, and it is consistent with the palaeogeographical reconstruction of this island. We think that this is strong evidence for our view that there was a lake or lake system there until Pliocene. We added this information in the revised manuscript (L339–342).

2) The assumption that the fossil “…†Lithopoecilus brouweri is the common ancestor of O. sarasinorum and O. eversi…” (l. 335-336) is likewise pure speculation. The morphology of this fossil fish remains largely unclear: Frickhinger 1991 (cited here) illustrates the fossil, but gives little more information than that this is a small and slender fish with large eyes and a pointed head; Parenti 2008 (the second source cited here) says that she did not see the specimen, but that she sees no reason to contradict Beaufort 1934, who said that it is intermediate between Oryzias and Adrianichthys. In sum, the knowledge of †Lithopoecilus brouweri morphology is very limited. It can be said that a ricefish that was considered in 1934 being similar to other Sulawesi ricefishes occurred in the area in the Miocene – that’s it. Anything else would require new studies of this fossil specimen.

Parenti (2008) stated that “I have not examined the fossil and cannot place it unambiguously in either Oryzias or Adrianichthys (p.592)”, implying that she accepted the Beaufort’s observation that †Lithopoecilus is intermediate between the two genera. Either way, we weakened our argument by stating “we think that †L. brouweride is the common ancestor of O. sarasinorum and O. eversi, but further examinations of this fossil species are necessary” (L337–339).

Further, I do not share the view that tiny Tilanga fountain is an environment that is to be termed “lacustrine”. Lacustrine refers to a “lake environment”, whereas the Tilanga fountain is a tiny pool, connected to groundwater, with a small overflow. The point I raise is however less that of the terminology, it questions if the habitat is from its properties somehow comparable to “real” lakes, such as Lake Lindu: Substantial and long-lived, largely stagnant waters. To my understanding, there is also no evidence that the Tilanga fountain can be seen as a leftover of a lake – its simply a fountain, and we do not know more about that habitat so far.

Because our point in this study is that a large ancient lake(s) in central Sulawesi was fragmented into several small ones, we would like to keep the term “lacustrine” in this manuscript. So, we decided to compromise with the referee’s point as follows. First, we deleted the definition of “lacustrine” from the materials and methods section (L122) and used “lacustrine and/or pond” when the mention there was limited only to O. sarasinorum and O. eversi (L50 and L299). In contrast, we used “lacustrine” where our mention was about adrianichthyids or other lacustrine organisms in central Sulawesi in general (L52, L73, L97, L98, L312, L321, and L368). We would like to keep “lacustrine” in the title, because this title includes our message as above.

Having raised these points of criticism, I would like to express that I would in fact like seeing these interesting genetic results published, but with an interpretation that is based on the existing body of evidence. I feel that the idea of interconnection of the two sites, Tilanga and Lindu, by a lake or several lakes, could be proposed as one possible scenario – stating clearly that there is so far no further evidence available. And I would expect a critical discussion, including alternative scenarios. If this is done, the present paper will also remain valid if it should turn out later, that that there was in fact no lake connection among Tilanga and Lindu.

In this study, we compared two alternative scenarios, i.e., incomplete lineage sorting (ILS) and ancient admixture, using model-based genetic analyses. The results clearly supported the ancient admixture scenario rather than ILS. Although we agree that geological studies would be needed to demonstrate our hypothesis, the existence of †Lithopoecilus brouweri seems to be strong enough to support our hypothesis for the moment. In the present circumstance, there is no alternative scenario or hypothesis available to explain our results, and even if any, there would be no foundation.

The same applies to the Tilanga habitat: Terming this “lacustrine” raises the expectation that these are two populations of lake fish, which is not the case. Saying openly that this is a “fountain” would appear to me even more interesting, and clearly correct.

As above, we used “lacustrine and/or pond” when our mention was limited to O. sarasinorum and O. eversi (L50 and L299).

---

## [Decision Letter · Decision Letter 2]

6 May 2021

Mitochondrial introgression by ancient admixture between two distant lacustrine fishes in Sulawesi Island

PONE-D-20-40353R2

Dear Dr. Yamahira,

We’re pleased to inform you that your manuscript has been judged scientifically suitable for publication and will be formally accepted for publication once it meets all outstanding technical requirements.

Kind regards,

Tzen-Yuh Chiang

Academic Editor

PLOS ONE

Additional Editor Comments (optional):

Reviewers' comments:

Reviewer's Responses to Questions

**Comments to the Author**

1. If the authors have adequately addressed your comments raised in a previous round of review and you feel that this manuscript is now acceptable for publication, you may indicate that here to bypass the “Comments to the Author” section, enter your conflict of interest statement in the “Confidential to Editor” section, and submit your "Accept" recommendation.

Reviewer #1: All comments have been addressed

Reviewer #3: All comments have been addressed

2. Is the manuscript technically sound, and do the data support the conclusions?

Reviewer #1: Yes

Reviewer #3: Yes

3. Has the statistical analysis been performed appropriately and rigorously? 

Reviewer #1: Yes

Reviewer #3: Yes

4. Have the authors made all data underlying the findings in their manuscript fully available?

Reviewer #1: Yes

Reviewer #3: Yes

5. Is the manuscript presented in an intelligible fashion and written in standard English?

Reviewer #1: (No Response)

Reviewer #3: Yes

6. Review Comments to the Author

Reviewer #1: (No Response)

Reviewer #3: Reviewer's report

Date: May 6, 2021

Journal: PLOS ONE

Manuscript Number: PONE-D-20-40353R2

Title: "Mitochondrial introgression by ancient admixture between two distant lacustrine fishes in Sulawesi Island"

Authors: Horoiwa et al.

The authors have accomplished a genome-wide analyses of putative historical hybridization and introgression between two ricefishes (family Adrianichthyidae), Oryzias eversi and O. sarasinorum.

The manuscript is clear and well written, with no fundamental flaws and weaknesses, and contains new and interesting data that are sound, adequately described and illustrated, and that may provide important cues to scientists interested in thereby support the usage of nuclear and mitochondrial sequences in evolutionary studies. Therefore the manuscript is suitable for publication in PLOS ONE.

Minor point:

The RDP program (Martin et al., 2015) detects at least 15 recombination events within the 10 complete mt genomes of Oryzias celebensis, O. dancena, O. javanicus, O. latipes, O. luzonensis, O. melastigma, O. minutillus, O. sarasinorum, O. sinensis, and O. curvinotus (GenBank data). It means that admixture and historical introgression might be frequent for these fishes. The analysis and conclusions of this manuscript could be more comprehensive with the additional data on complete mitochondrial genomes.

References

Martin DP, Murrell B, Golden M, Khoosal A, & Muhire B (2015) RDP4: Detection and analysis of recombination patterns in virus genomes. Virus Evolution 1: vev003 doi: 10.1093/ve/vev003

7. PLOS authors have the option to publish the peer review history of their article (what does this mean?). If published, this will include your full peer review and any attached files.

Reviewer #1: No

Reviewer #3: No

---

## [Editor Report · Acceptance letter]

31 May 2021

PONE-D-20-40353R2 

Mitochondrial introgression by ancient admixture between two distant lacustrine fishes in Sulawesi Island 

Dear Dr. Yamahira:

I'm pleased to inform you that your manuscript has been deemed suitable for publication in PLOS ONE. Congratulations! Your manuscript is now with our production department. 

Kind regards, 

on behalf of

Dr. Tzen-Yuh Chiang 

Academic Editor

PLOS ONE